# An Upper Ocean Thermal Field Metrics Dataset

Charles R. Sampson [1,*], James Cummings [2], John A. Knaff [3], Mark DeMaria [4] and Efren A. Serra [5]

1 Naval Research Laboratory, Monterey, CA 93943, USA
2 Science Applications International Corporation, Naval Research Laboratory, Monterey, CA 93943, USA
3 NOAA Center for Satellite Applications and Research, Fort Collins, CO 80523-1375, USA
4 Cooperative Institute for Research in the Atmosphere, Colorado State University, Fort Collins, CO 80523-1375, USA
5 Devine Consulting, Naval Research Laboratory, Monterey, CA 93943, USA
* Correspondence: buck.sampson@nrlmry.navy.mil

**Abstract:** The upper ocean provides a source of thermal energy for tropical cyclone development and maintenance through a series of complex interactions. In this work, we develop a seventeen-year dataset of upper ocean thermal field metrics for use in tropical cyclone studies and development of tropical cyclone intensity prediction models. These metrics include the surface temperature, two different measures of vertically integrated heat content, and four different measures of vertically averaged temperature. Some metrics have been used to study upper-ocean energy response to tropical cyclone passage, while others have been employed to improve operational tropical cyclone intensity prediction models. The vertically integrated ocean heat content has been used to improve tropical cyclone intensity forecasts at U.S. tropical cyclone forecast centers and is an integral part of several operational intensity forecast models. A static 2005–2021 dataset that includes all twelve metrics described within is available on the Naval Research Laboratory web server, and a subset of six metrics have been produced in real-time at Fleet Numerical Meteorology and Oceanography Center and provided to the public via the GODAE server since 2021.

**Keywords:** upper ocean; tropical cyclones; NCODA; rapid intensification; intensity change; SST

## 1. Introduction

The ocean and the atmosphere form a complex coupled system in which heat is stored, transported and exchanged. The effect of the ocean as a source of thermal energy for the overlying atmosphere can be considerable, leading to effects such as sea breezes, tropical cyclone (TC) formation and intensification, and larger-scale effects such as El Niño/La Niña Southern Oscillation (ENSO), which causes thermal variations in the upper eastern Pacific Ocean that can lead to shifts in weather patterns over large spatial scales. Quantifying the heat content of the upper ocean is important in determining such air/sea interaction. The upper ocean is typically characterized by a layer that is largely homogeneous in temperature, salinity and density. This mixed layer is the source for heat and moisture fluxes to the atmosphere above, and so variations in ocean mixed layer parameters can greatly affect the overlying atmosphere. Therefore, it is no surprise that these variations in the ocean mixed layer parameters are important to atmospheric prediction and climate models. Extensive comparisons of mixed layer definitions have been completed using density profiles in studies such as [1].

The question that remains is how to define representative metrics to describe thermal properties of the mixed layer appropriate for tropical cyclone intensity analysis and forecasting. In this work, we present several ways to define the ocean mixed layer and several ways to quantify the thermal properties in that layer. The metrics employed here were suggested by [2]. We derive these metrics for a 17-year (2005–2021) time period over a 65N–65S band around the globe using ocean analysis fields.

In Section 2, we describe the data used for deriving the metrics. In Section 3, we define the metrics and present examples of the resulting two-dimensional fields. In Section 4, we summarize the results and discuss some potential applications for the new dataset.

## 2. Materials and Methods

Global ocean analyses are produced using the Navy Coupled Ocean Data Assimilation (NCODA) system run, upgraded, and maintained at the U.S. Navy Fleet Numerical Meteorology and Oceanography Center in Monterey, CA, U.S. and Stennis Space Center, MS, U.S. as an operational algorithm. The NCODA analyses have been produced since June 2005 to the present time. From 2005 to 2013 NCODA used a Multi-Variate Optimal Interpolation (MVOI) scheme [3]. Since 2014 NCODA has used a multivariate three-dimensional variational (3DVAR) method [4]. In analysis-only mode NCODA uses the prior ocean analysis as a first guess. By avoiding use of a numerical model first-guess, NCODA analyses are independent of ocean model and atmospheric model forcing errors, especially those associated with the physical parameterization of mixing. However, the quality of the analysis can be affected by lack and latency of observations. As such, NCODA maintains an analysis variable that estimates the age of the observations on the grid. In the case of grid locations that have not been influenced by observations for more than 30 days, profiles from the monthly Navy Generalized Digital Environmental Model ocean climatology are introduced into the analysis as synthetic observations. The purpose here is to ensure that the analysis-only system maintains a seasonal cycle.

NCODA analyses are a result of an observational data fitting approach to the previous analysis, used here as a persistence forecast. Prior to 2013 NCODA used a 24 h update cycle (a data time window of ±12 h centered around the analysis time), while since that time NCODA has used a 12 h update cycle (a data time window of ±6 h). Conventional observational data for the analyses are obtained from the Global Telecommunications System (GTS, https://public.wmo.int/en/programmes/global-telecommunication-system, accessed on 22 August 2022), with satellite data obtained directly from the data providers. All data assimilated are subject to ocean data quality control (QC) procedures [5], and are made available, with QC outcomes, on the U.S. Global Ocean Data Assimilation Experiment (GODAE) data server (https://www.usgodae.org, accessed on 22 August 2022). NCODA assimilates satellite altimeter sea surface height anomaly (SSHA) observations, satellite and in situ sea surface temperatures (SST), as well as available in situ vertical temperature and salinity profiles from XBTs, Argo floats, moored buoys, ocean gliders, and ship-board CTDs. Note that the SSHA data are assimilated along-tracks and are first converted to temperature and salinity profiles using historical relationships between dynamic height and temperature at depth. Salinity is then derived from the estimated temperatures using temperature-salinity correlations that vary with depth, time-of-year, and location [6].

NCODA produces three-dimensional analyses of temperature and salinity, from which geopotential (dynamic height) and geostrophic velocity are derived. Since April, 2006 the analysis has been calculated on a 1/6-degree resolution grid with 34 vertical levels using a stretched vertical grid ranging from 0 m to 5000 m depth. There are 16 levels defined in the upper 400 m of the water column. Prior to April 2006 the analysis used a 1/4-degree resolution grid with 32 vertical levels. For this study, we use archived NCODA analyses from 2005–2021 to generate a suite of two-dimensional fields of the metrics described in the following section. These derived products span the globe between 65° N and 65° S and are produced on a cylindrical grid at 0.25° resolution.

## 3. Results

### 3.1. Derived Metrics

In this section we present seven approaches to quantifying upper ocean structure and heat content. Example grids of each metric will be presented. Graphical examples of these derived products are also produced when applicable. To facilitate comparison, data from 15 September 2010 are used in all examples presented here.

### 3.1.1. Sea Surface Temperature

The simplest metric is the Sea Surface Temperature (SST), the temperature of the top of the mixed layer. This metric has proven useful in determining conditions favorable for tropical cyclone (TC) formation and intensification. TC potential intensity is a parameter determined empirically as a function of SST [7–11]. Many empirical and theoretical models of potential intensity also have been developed (e.g., [12–15] and other references contained therein), which further highlight the importance of the ocean as the ultimate TC energy source. The theoretical models also include the influence of the atmosphere on the potential intensity, but under most circumstances, the ocean influence is comparable or greater than that of the atmosphere.

An example SST grid valid on 15 September 2010 is presented in Figure 1. The tropics and northern hemisphere oceans in mid-September exhibit large areas of warm SST, with some regions in the western North Atlantic, Pacific and Indian Ocean basins having temperatures exceeding 26 °C. Many northern hemisphere tropical cyclones form and intensify in these regions.

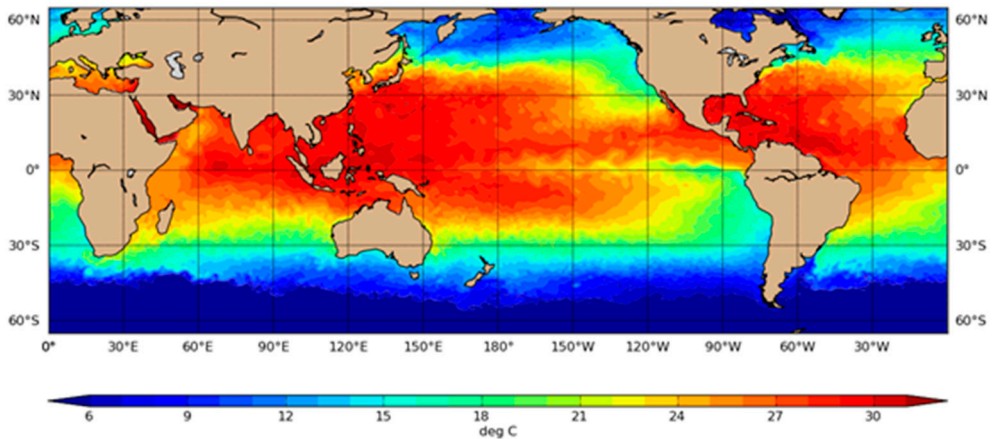

**Figure 1.** SST (°C) on the 0.25° cylindrical grid, valid 15 September 2010.

### 3.1.2. Ocean Heat Content Using 26 °C Isotherm (OHC26)

A quantification of upper oceanic heat content for TC development was first presented by [16] as the integrated temperature in excess of 26 °C isotherm (the commonly agreed upon lower limit for TC development) from the depth of the 26 °C isotherm ($Z_{26\,°C}$) to the surface (0), which we will referred to as Oceanic Heat Content (OHC) (Emanuel [14] called this quantity "hurricane heat potential". This quantity has also been referred to as "tropical cyclone heat potential" as in [15]) defined by

$$\text{OHC}(x,y) = \rho_o C_P \int_{-Z_{26°C}}^{0} [T(x,y,z) - 26] dz \tag{1}$$

where $\rho_0$ = 1026 kg m$^{-3}$ and $C_p$ = 4187 J kg$^{-1}$ are the mean density and heat capacity assigned for water, respectively. OHC has been used in a variety of TC research and operational applications as reviewed in [17].

One disadvantage of this definition of OHC is that the 26 °C isotherm outcrops in cooler ocean water, which leads to areas where OHC is undefined. In the 15 September 2010 example (Figure 2) this outcropping generally occurs in regions where TCs decay, but equatorward of that often preferred for TC studies and model development. The topology depicted in Figure 2 shows that in some regions the warm surface conditions extend to different depths that may not be obvious from considering only the surface data.

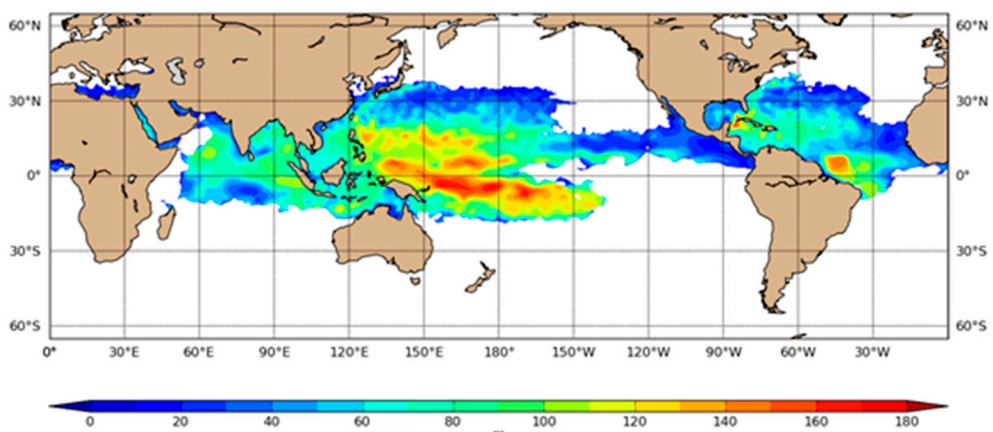

**Figure 2.** Depth (m) of the 26 °C isotherm on 15 September 2010.

To provide a continuous value of OHC over the entire analysis region, we use a slightly different definition of the heat content of the ocean in those regions where the isotherm outcropping occurs. We use the same vertical integral as Equation (1), but applied over a layer defined as the level where the temperature difference from the surface is less than 1.0 °C.

$$OHC(x, y) = \rho_o C_P \int_{-Z_m}^{0} [T(x, y, z) - 26] dz$$

where $z_m$ is the depth where the temperature difference from the surface is less than 1 °C, $\rho_0 = 1026$ kg m$^{-3}$ and $C_p = 4187$ J kg$^{-1}$ are the mean density and heat capacity assigned for water, respectively.

Because the sea water temperatures are less than the 26 °C reference temperature in Equation (1), the heat content has negative values relative to sea water in regions above that temperature. In very cold water we limit the negative heat values to −240 kJ cm$^{-2}$ for display purposes only.

Figure 3 depicts the OHC with negative values for the 15 September 2010 data. Most of the wintertime southern hemisphere oceans have negative heat content. The western parts of the North Pacific and North Atlantic have large regions of high OHC26, indicating areas where the heat content is greater due to both increased SST and deeper 26 °C isotherm levels. There are also significant areas where the OHC26 is between 0 and −100 kJ cm$^{-2}$, and these negative values could provide additional and spatially continuous information to TC model developers.

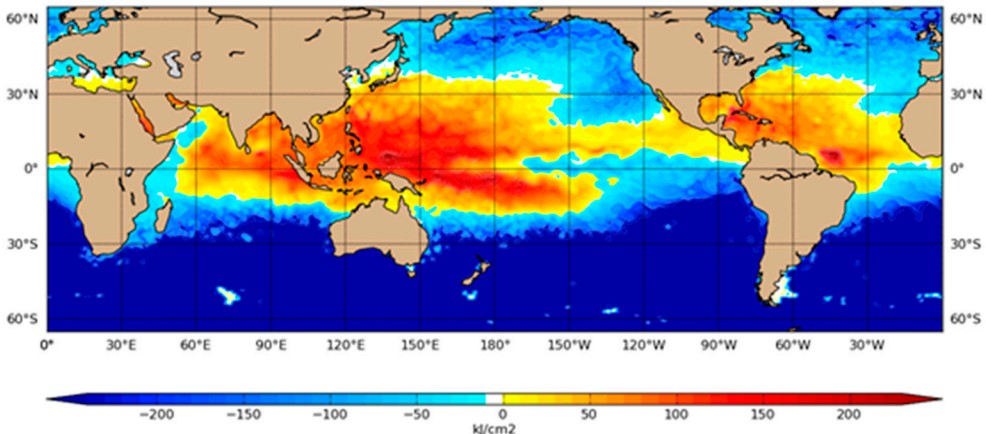

**Figure 3.** Ocean heat content (kJ cm$^{-2}$) integrated from the surface down to the 26 °C isotherm (OHC26) on 15 September 2010.

### 3.1.3. Ocean Heat Content Using 20 °C Isotherm (OHC20)

Another way to compute OHC is to apply the vertical integral to the 20 °C isotherm (OHC20). This has the effect of deepening the layer over which upper ocean heat is considered to be available for interaction with the atmosphere. This could provide useful information in areas where surface stress-induced mixing might apply to deeper levels, such as under an intense atmospheric storm. The 20 °C isotherm is embedded in the permanent thermocline of the tropical ocean. As such it is less likely to be influenced by local heating and cooling than integrals computed using the 26 °C isotherm. The alternate definition also expands the area of inclusion before outcropping occurs (compare Figure 4 to Figure 2). As with OHC26, our definition of OHC20 includes negative heat values.

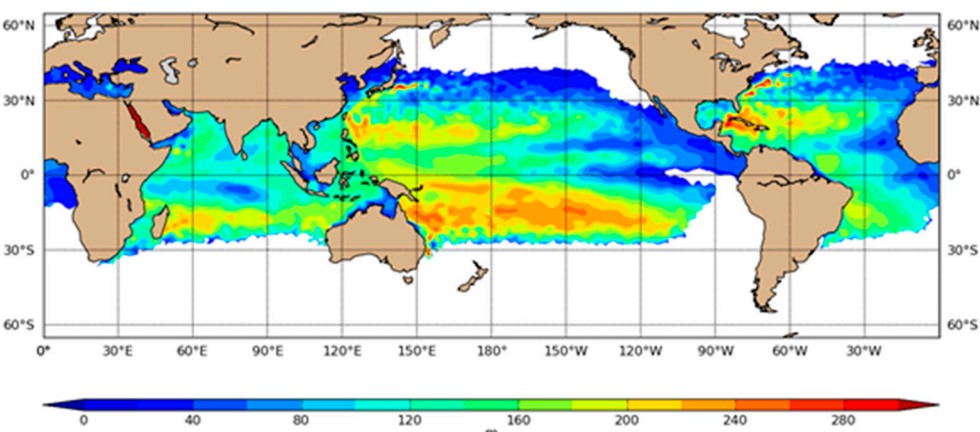

**Figure 4.** Depth (m) of the 20 °C isotherm on 15 September 2010.

Figure 5 depicts the same case as in Figure 3 except for OHC20. The high heat content areas are similar among the two approaches although the extent of positive OHC is larger in OHC20 than in OHC26 (as expected). A difference plot (OHC20-OHC26) for this specific case is included in Appendix A.

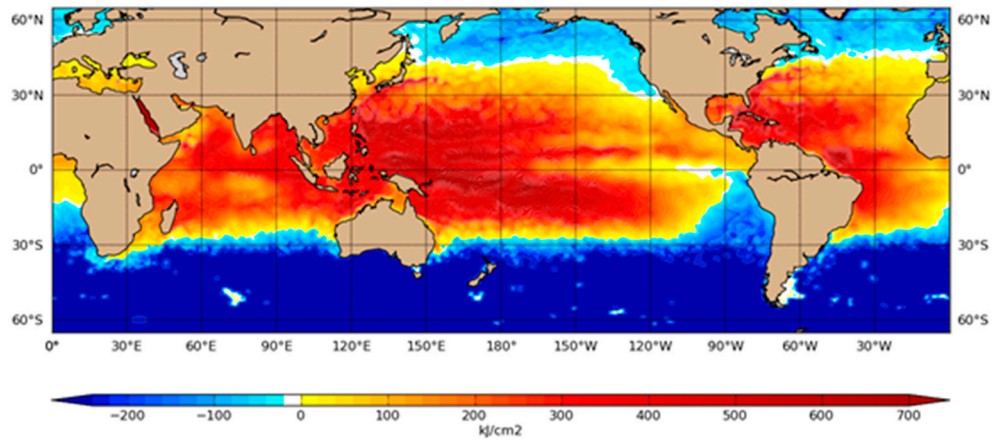

**Figure 5.** Ocean heat content (kJ cm$^{-2}$) integrated from the surface down to the to 20 °C isotherm (OHC20) on 15 September 2010.

### 3.1.4. Average Temperature to 100 m (T100)

The OHC defined in Sections 3.1.2 and 3.1.3 has several shortcomings, as pointed out by [2]. As originally defined, it is limited to regions where the reference isotherm does not outcrop. In shallower waters, the ocean may not be deep enough to include the reference SST isotherm, causing a potential misrepresentation of the ocean conditions. Finally, OHC does not address static stability changes with depth in salt-stratified waters.

Price [2] proposes that a more relevant measure of upper oceanic energy may be obtained from an average upper ocean temperature as defined by

$$\overline{T_d}(x,y) = \frac{1}{d}\int_{-d}^{0} T(x,y,z)dz \qquad (2)$$

where the *d* is the depth of vertical mixing caused by a TC. Price [2] further described two ways to define the mixing depth *d* in Equation (2). The first assumes that the typical mixing depth associated with a mature TC passage is 100 m—a simple yet realistic assumption, and the second calculates the mixing depth directly from the ocean temperature profile. T100 is simple to calculate and understand, and provides a continuous measure of upper ocean heat even in relatively shallow water (e.g., in the shallows of the Gulf of Mexico where the entire water column is warmer than 26 °C).

When applied to our 15 September 2010 case (Figure 6), T100 provides a heat estimate that is representative of a layer rather than SST (Figure 1).

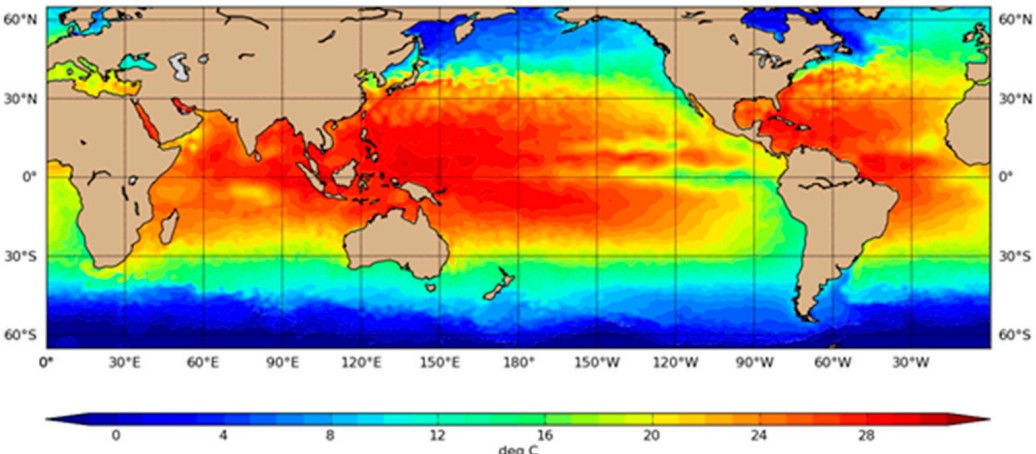

**Figure 6.** Layer-weighted mean temperature (°C) to a depth of 100 m (T100) on 15 September 2010.

3.1.5. Average Temperature to Temperature Difference Mixed Layer Depth (Td_ΔT_0.5)

Price's [2] second suggested method is to determine d in Equation (2) from the ocean pro-file. The three remaining heat metrics in this study use different approaches to select the depth *d*. In the next three sections we select depths *d* to represent the level to which atmospheric interaction with the ocean water occurs.

The mixed layer is separated from the thermocline below by the barrier layer leading to stable stratification. The heat contained in the mixed layer is more available to the atmosphere than that contained in the stable layer below. Here, we use the approach of [17] which defined the mixed layer as the depth at which the temperature change (positive or negative) from the surface temperature is 0.5 °C.

For the 15 September 2010 case, the mixed layer depth as defined by [17] is depicted in Figure 7. This date is at the end of the austral winter when months of cooling due to decreased solar heating and mixing from wind-driven turbulence have led to much larger mixed layer depths in the southern hemisphere mid-latitudes. In the northern hemisphere (boreal summer) there is more stable stratification leading to typical mixed layer depths that are much shallower than the 100 m chosen by [2].

The mean temperature of the mixed layer using a temperature difference of 0.5 °C is depicted in Figure 8. Comparing to T100 (Figure 6) there are larger regions of greater Td_0.5 values over the western North Pacific and Atlantic oceans where surface temperatures are high, indicating the average is calculated over a shallower layer.

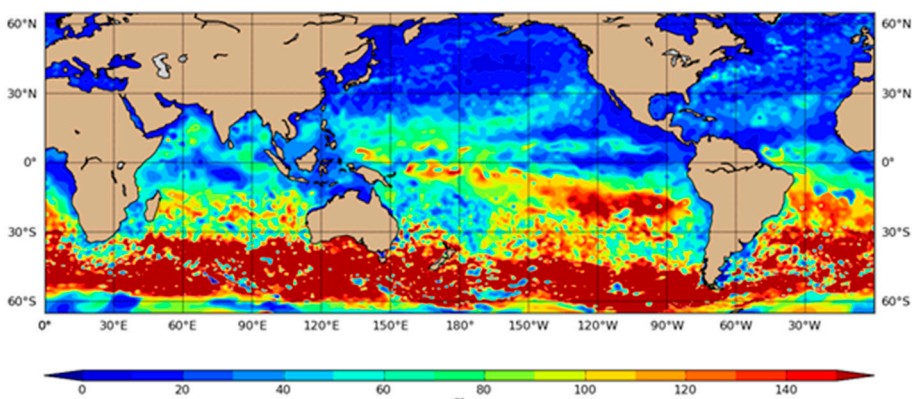

**Figure 7.** Mixed layer depth (m) defined by temperature difference from surface of 0.5 °C on 15 September 2010.

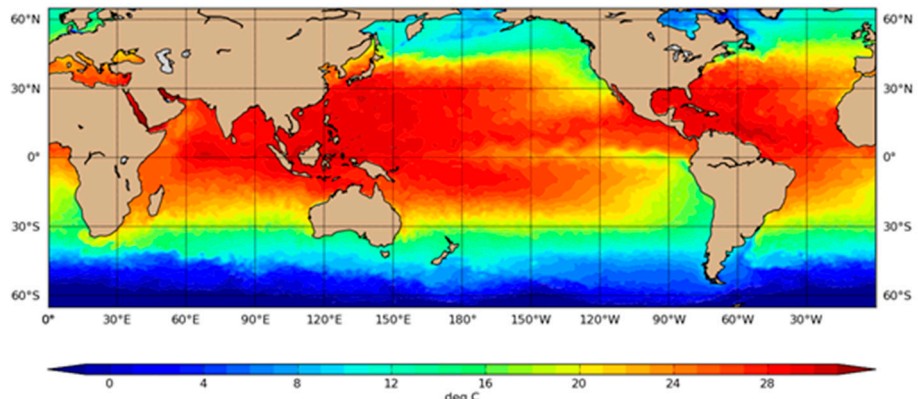

**Figure 8.** Layer-weighted mean temperature (°C) to mixed layer depth defined by ΔT = 0.5 °C on 15 September 2010.

### 3.1.6. Average Temperature to Potential Density Difference Mixed Layer Depth (Td_ρθ_0.15)

Another approach to defining the mixed layer depth is to use the upper ocean potential density, the density that a parcel would acquire if brought adiabatically to the surface. For the mixed layer to be statically stable, potential density increases with depth. We define the mixed layer depth as the level where potential density increases by 0.15 kg m$^{-3}$. In the example Td_ρθ_0.15 topology (Figure 9) and metric field (Figure 10) it can be seen that the results are similar to those using Td_ΔT_0.5 (Figures 7 and 8, respectively).

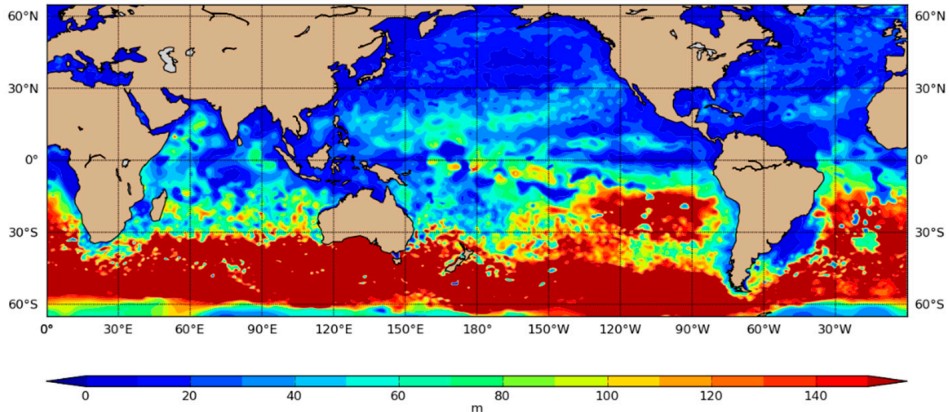

**Figure 9.** Mixed layer depth (m) defined by potential density difference of 0.15 kg m$^{-3}$ from surface on 15 September 2010.

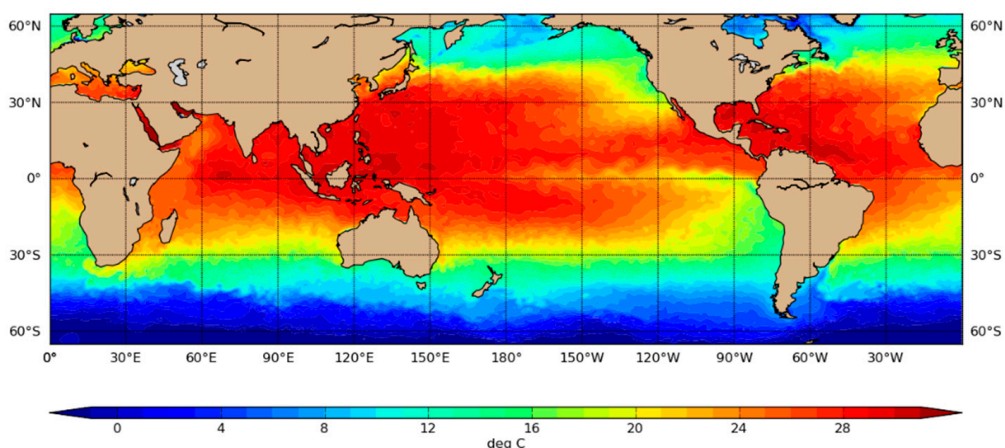

**Figure 10.** Layer-weighted mean temperature (°C) to mixed layer depth defined by $\Delta\rho\theta = 0.15 \text{ kgm}^{-3}$ from surface on 15 September 2010.

3.1.7. Average Temperature to Level of Maximum Stability (Td_MaxE)

In this metric we define the depth of the ocean barrier layer using the ocean stability (E) and Equation:

$$E = -1/\rho \cdot (d\rho/dz) \tag{3}$$

where ρ is the density of sea water. Stability is defined such that

E > 0 Stable
E = 0 Neutral Stability
E < 0 Unstable

Thus, we calculate E at each depth of the upper ocean profile and choose the maximum stability to indicate the barrier depth. As can be seen in Figure 11, over much of the oceans the depth of maximum stability is relatively shallow and discontinuous. This is likely not a desired quality for a tropical cyclone related metric. There are some areas of deeper maximum E, mainly in the North Pacific Ocean tropics, but much of the field is near zero. The resultant Td_MaxE metric are depicted in Figure 12, and looks much more continuous despite the maximum E issues.

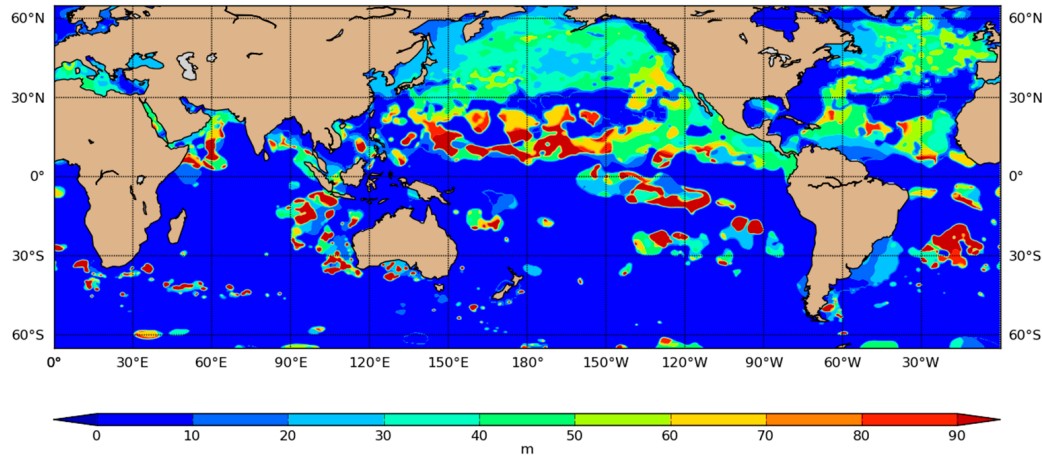

**Figure 11.** Depth of maximum stability (m) on 15 September 2010.

Difference plots between the mean temperatures computed for the estimates of mixed layer depth and T100 are shown in Appendix A. The average temperatures for the layers computed dynamically are well correlated, but they are all quite different than the mean temperatures computed through 100 m depth (T100).

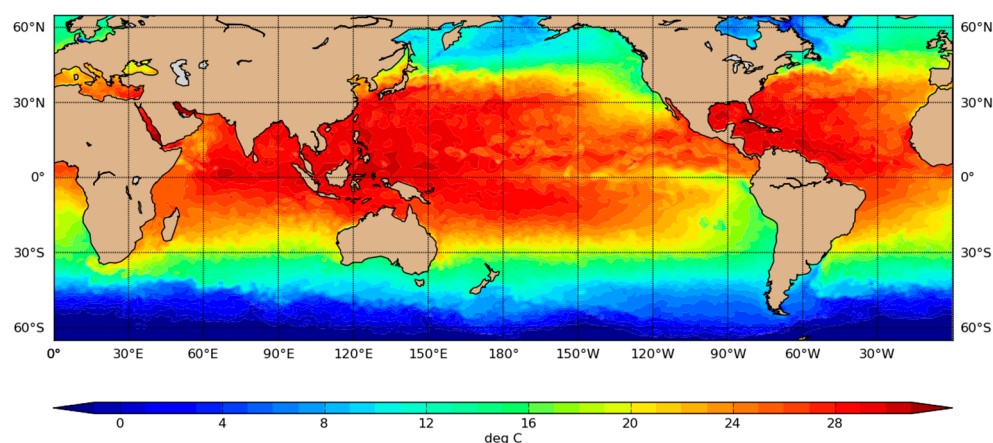

**Figure 12.** Layer-weighted mean temperature (°C) to the depth of maximum stability on 15 September 2010.

### 3.2. Tropical Cyclone Applications Using Upper Ocean Thermal Metrics

The importance of the upper ocean as a boundary condition for tropical cyclones is highlighted in [12–15] leading to theoretical [13,14] and empirical [7–11,15] estimates of tropical cyclone potential intensity based on the SSTs that exist under the tropical cyclone eyewall. In a real-time forecasting environment, the SST information available for this quantity is usually based on observations made prior to the storm's arrival (i.e., via satellite based techniques). However, tropical cyclone winds act to mechanically mix the upper ocean as they arrive on the scene as described in [2], the prior SSTs can be significantly modified, and this mixing action predominantly results in cooler SST conditions near the storm's center. To account for this cooling both empirical and numerical models that make future intensity forecasts use quantities in the upper ocean to account for the ocean mixing.

The metrics developed in this paper, specifically the OHC26, have been used to account for the ocean mechanical mixing and the reduction in pre-storm SSTs. When the OHC26 is large and the mixed layer depth both deep and less stably stratified, mixing results in less in-storm SST cooling (e.g., [18–21]). Higher values of OHC26 also provide more favorable conditions for rapid intensification (defined as 95th percentile of intensity changes) when the rest of the TC environment is also favorable (e.g., [22,23]).

For this reason, statistical-dynamical tropical cyclone intensity models have incorporated OHC26 as predictors, which have been shown to improve intensity forecasts. Examples of uses of OHC26 in operational models include the Statistical Hurricane Prediction Scheme (SHIPS [24]), and the Logistic Growth Equation Model (LGEM [25]). These models are used in operations at NOAA National Hurricane Center, the Central Pacific Hurricane Center, and U.S. Department of Defense's Joint Typhoon Warning Center, producing skillful (vs. climatology + persistence baseline models) intensity forecasts.

One of the more challenging tropical cyclone intensity forecasts are those involving rapid intensification, which typically occurs in most tropical cyclones reaching 50 m s$^{-1}$ maximum sustained 1 min wind speeds [26]. To address these forecasts several probabilistic schemes are run in U.S. operational tropical cyclone forecasts. These include the SHIPS Rapid Intensification Index and Rapid intensification consensus [22], the Rapid Intensification Prediction Aid [23,27] and the Forest Rapid Intensification Aid (FRIA [28]). Each of these models uses OHC26 as a predictor. OHC26 typically affects predictors related to the potential intensification, short-term intensity trends, and vertical wind shear predictors. Although these models are constructed differently, OHC increases their reliability and accuracy (e.g., Pierce and Brier skill scores) in rapid intensity probability prediction.

Diagnostic studies have also been carried out using composite analyses of these metrics. Specifically, the response of the upper ocean to tropical cyclone passage was studied in [29]. Findings of that study suggest that upper oceanic energy decreases in these metrics are shown to persist for at least 30 days—long enough to possibly affect future TCs,

and indicates that tropical cyclone kinetic energy (KE) should be considered when assessing TC impacts on the upper ocean. In addition, SST changes are best related to the KE and the latitude whereas the upper ocean energy changes are a function of KE, initial upper ocean energy conditions, and translation speed. Finally, the "typical" tropical cyclones at 30° latitude 10-day lagged SST cooling is approximately 0.78 °C whereas the same storm results in 10-day and 30-day lagged decreases in OHC26 by about 12 and 7 kJ cm$^{-2}$ and a 0.58 °C and 0.38 °C cooling in T100 metrics, respectively.

Thermal parameters seen in Figures Figures 1–5 and 7 are generated in real-time operations at the U.S. Navy Fleet Numerical Meteorology and Oceanography Center and are available from the GODAE server (https://usgodae.org/index.html, accessed on 22 August 2022), web pages (e.g., https://rammb-slider.cira.colostate.edu/, accessed on 22 August 2022) and forecasts systems (see Figure 13).

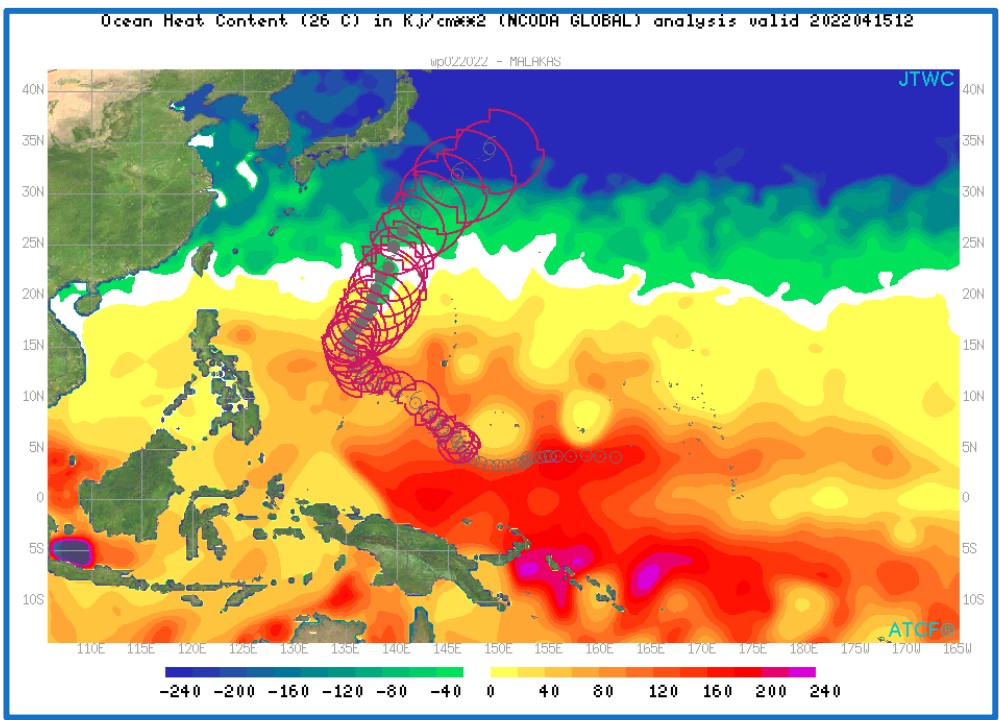

**Figure 13.** NCODA OHC26 (kJ cm$^{-2}$) real-time product on 15 April 2022 at 12:00 UTC displayed via the Automated Tropical Cyclone Forecast System. Typhoon Malakas (the second tropical cyclone of the 2022 western North Pacific season) is overlain as it decays over cooler water with negative OHC26 values off the coast of Japan.

## 4. Discussion

The metrics defined in the previous section have been derived using NCODA analyses produced by the Fleet Numerical Meteorology and Oceanography Center (FNMOC). NCODA is run twice daily (00 and 12 UTC) in operations, and fields are then posted for public use on the GODAE server (https://usgodae.org/index.html, accessed on 22 August 2022). The GODAE server only retains a subset of the parameters described in this work. At the time of this report, six of the variables discussed here are available in near real-time at ftp://www.usgodae.org/ftp/outgoing/fnmoc/models/glb_ocn/grib/ (accessed on 22 August 2022), and an example of a run is shown in Figure 14. The GODAE GRIdded Binary or General Regularly distributed Information in Binary form (GRIB) format repository of all six metrics extends from 2020 through the writing of this manuscript while three of the metrics (SST, OHC26 and Depth of 26 °C Isotherm) extend back through 2014.

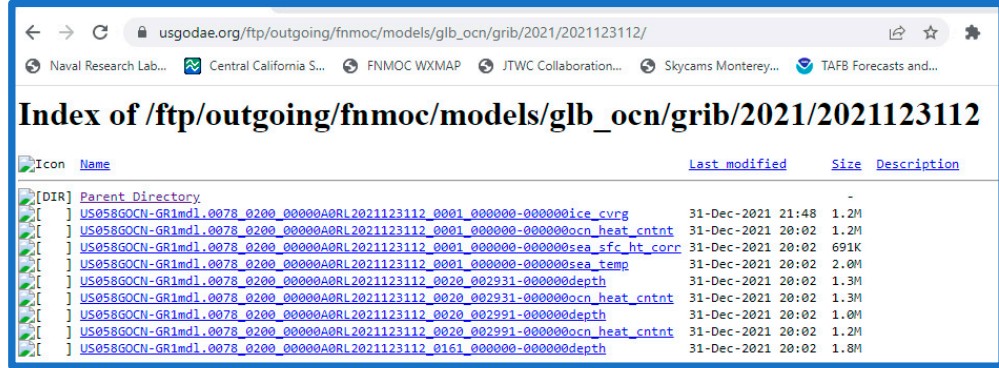

**Figure 14.** Upper Ocean Metrics available in real-time on the GODAE server. Metrics are global and include from top to bottom: Ice Coverage (not discussed), OHC26, Sea Surface Height Correction (not discussed), SST, Depth of the 20 °C Isotherm, OHC20, Depth of the 26 °C Isotherm, OHC26 (a second copy), and Mixed Layer Depth (5 °C).

A complete set of the metrics defined in Section 3 has been produced for research. These fields can be downloaded from the Naval Research Laboratory Marine Meteorology Division: http://www.nrlmry.navy.mil/atcf_web/nopp_ohc/ (accessed on 22 August 2022). An example of a FORTRAN reader for the data is also provided, as well as graphical images of parameters for approximately 10 of the years.

As discussed the horizontal and vertical grid configuration for this dataset has changed since NCODA was first implemented. These grid changes are not expected to have modified any of the results or conclusions presented in this paper.

The derived thermal metrics described here have already been used in a study of the ocean response to TC passage [30]. The results showed that passage of an average-sized, hurricane-strength TC results in typical SST cooling on the order of 0.6 °C which persists for about 30 days. The OHC26 is decreased by about 12 kJ cm$^{-2}$ and the T100 is cooled by about 0.5 °C. These upper-ocean energy decreases were shown to persist for up to 60 days. There are ongoing annual efforts to update operational TC models such as the SHIPS [24] and LGEM [25] use these metrics as inputs to improve estimates of potential intensity and/or potential intensification.

NCODA continues to evolve and improve as well. New ocean data sources continue to be added to the mix of observations used in the assimilation, including satellite derived sea surface salinity and ocean gliders that are often deployed as targeted observing systems in hurricane reconnaissance missions. It is now an important component of the U.S. Navy's operational global ocean forecast system (GOFS) that uses the Hybrid Ocean Circulation Model (HYCOM) as the forward model [30]. Finally, NCODA has recently been ported to the NOAA Environmental Model Center (EMC) as the ocean data assimilation component part of the Real-Time Ocean Forecast System (RTOFS, see polar.ncep.noaa.gov for more information, accessed on 22 August 2022). RTOFS currently uses HYCOM but will eventually use the Modular Ocean Model (MOM).

## 5. Conclusions

The thermal structure of the upper ocean affects how and how much energy the ocean fluxes to tropical cyclone, which affects the tropical cyclones' future development and maintenance through a series of complex interactions. In this work, we have described the development a seventeen-year dataset of upper ocean thermal field metrics for use in tropical cyclone and other studies. The metrics include the surface temperature, two different measures of vertically integrated heat content, and four different measures of vertically averaged temperature. These metrics have been used to study upper ocean energy response to tropical cyclone passage while others have been employed to improve tropical cyclone intensity prediction models. The vertically integrated ocean heat content

in particular has been used to improve tropical cyclone intensity forecasts at U.S. tropical cyclone forecast centers.

Future work will concentrate on the creation of some additional upper ocean metrics, specifically metrics that use salinity and the Brunt–Väisälä frequency, and improving the use of other measures of ocean heat content, specifically T100 and Temperature to Level of Maximum Stability (Td_MaxE). Recently, it has become apparent that knowledge of the salinity structure of the upper ocean, and static stability variations strongly influence the SST seen by the hurricane eyewall and the fluxes of energy into the tropical cyclone [17,20]. T100 was highlighted in [2] and earlier works by the same author as an alternative to OHC26, which may better anticipate SST changes in intense tropical cyclones that typically mix the upper 100 m of the ocean. T100 may be particularly helpful in shallow water conditions and in waters colder than 26 °C—noting that the formulation for OHC26 described here allows for negative values in such conditions. Similarly, the more dynamic average Td_MaxE may prove useful guidance for anticipating evolution of SSTs under TC eyewalls, and could also be used as an alternative to a 100 m mixing depth used in T100. Finally, work will continue to develop improved techniques/algorithms to anticipate tropical cyclone intensity changes for operational forecasters, and ones that make better use of the discussed upper ocean metrics, salinity and the Brunt–Väisälä frequency.

**Author Contributions:** C.R.S., writing—original draft, visualization and data curation, software; J.C., conceptualization, methodology and writing—review and editing; J.A.K., writing—original draft, conceptualization, investigation; M.D., writing—review and editing, investigation, funding acquisition; E.A.S., investigation, software. All have contributed substantially to this work. All authors have read and agreed to the published version of the manuscript.

**Funding:** Publication of this work was graciously funded by the Office of Naval Research, Program Elements 0602435N and 0603207N. Support for John Knaff comes from NOAA base funding.

**Data Availability Statement:** The upper ocean metrics discussed in this paper can be downloaded from http://www.nrlmry.navy.mil/atcf_web/nopp_ohc/ (accessed on 22 August 2022), and the majority of the data assimilated and select parameters from this work are available real-time on the U.S. Global Ocean Data Assimilation Experiment (GODAE) data server (https://www.usgodae.org, accessed on 22 August 2022).

**Acknowledgments:** This article is dedicated to Jim Peak, retired NRL employee. Jim is the main contributor on the work and also wrote the original manuscript. We wish Jim all the best in retirement, wherever that may take him. The scientific results and conclusions, as well as any views or opinions expressed herein, are those of the author(s) and do not necessarily reflect those of NOAA or the Department of Commerce. We thank Michael Frost of NRL for keeping the GODAE server running all these years and his efforts to support operations at JTWC. We also thank Mark Ignaszewski for his tireless efforts at FNMOC to keep NCODA products running in operations, and Wayne Schubert for his groundbreaking work in the NOPP project.

**Conflicts of Interest:** The authors declare no conflict of interest.

## Appendix A

This appendix is dedicated to diagnosis of difference fields for both integrated temperatures and the two measures of ocean heat content (OHC26 and OHC20). A complete set of difference graphics for 2010 is included on the data repository website (http://www.nrlmry.navy.mil/atcf_web/nopp_ohc/, accessed on 22 August 2022), while a single case (15 September 2010) is shown in this appendix.

Differences for integrated temperatures fields based on the three different measures of mixed layer depth and a constant 100 m depth are shown in Figure A1. The gradients in differences tend to be small for the temperature averages calculated from three methods to measure mixed layer depth, but large differences exist between these mixed layer depths and the traditional in-TC measure of the 100- m depth (T100).

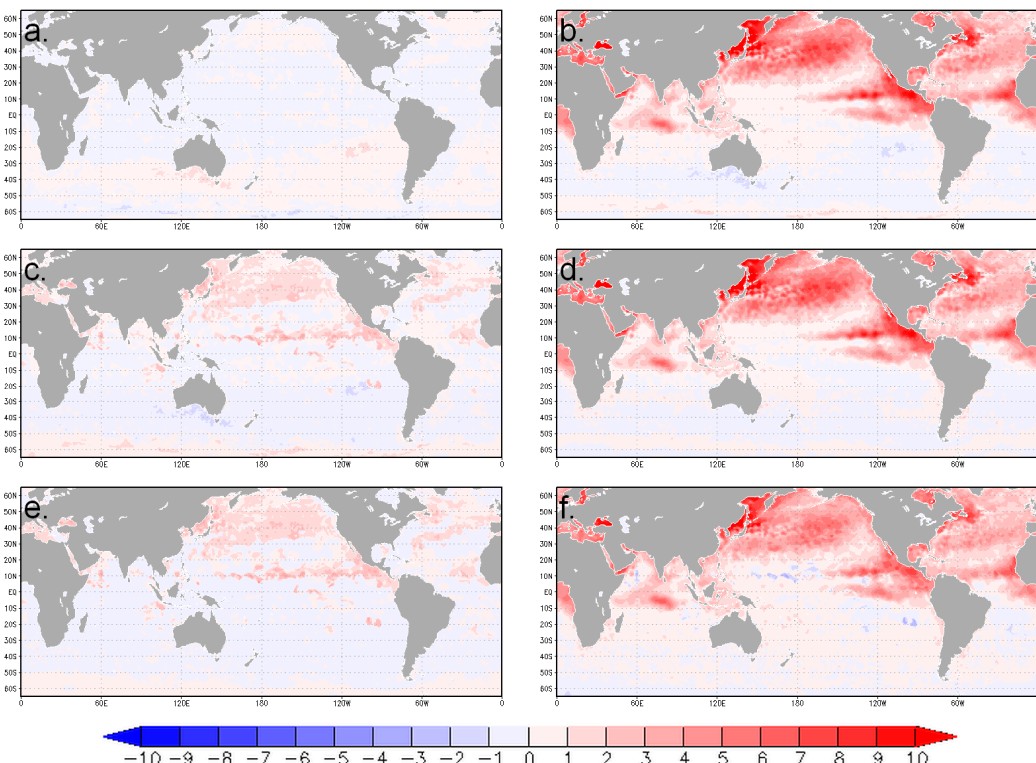

**Figure A1.** Differences between various measures of mean layer temperatures on 15 September 2010. (**a**) Td_ΔT_0.5- Td_ρθ_0.15, (**b**) Td_ρθ_0.15-T100, (**c**) Td_ρθ_0.15 -TD MaxE, (**d**) Td_ΔT_0.5-T100, (**e**) Td_ΔT_0.5-TD MaxE, (**f**) TD MaxE-T100.

Differences for the two measures of ocean heat content are shown in Figure A2. The OHC20 is always greater than or equal to OHC26 because of how they are defined. Differences tend to be largest in the tropics, especially in the Southwest Pacific. Differences in the Northern Hemisphere are largest off the East Coast of North America and in the deep tropics.

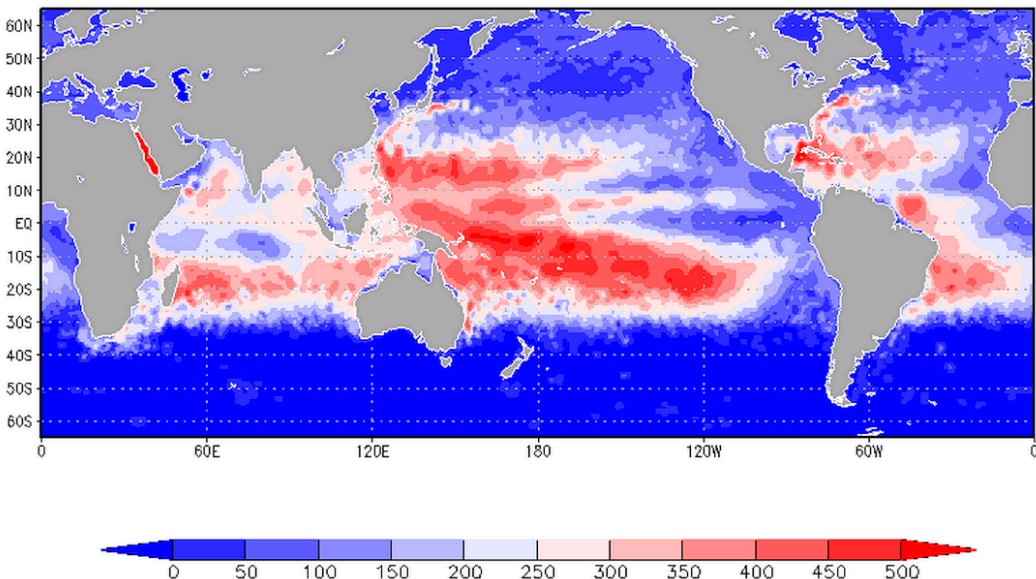

**Figure A2.** Difference between OHC20 and OHC26 fields on 15 September 2010.

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
