# Peer review of "An Upper Ocean Thermal Field Metrics Dataset"

_2674-0494, doi:10.3390/meteorology1030021_

Round 1

Reviewer 1 Report

Summary

The upper ocean provides a source of thermal energy for tropical cyclone development and maintenance. This study discussed the advantages and shortcomings of several metrics that represent the upper ocean thermal state. Using the NCODA three-dimensional oceanic analyses, six metrics, including two different measures of vertically-integrated heat content and four different measures of vertically-averaged temperature, were calculated and discussed. Finally, an upper ocean thermal field metrics dataset is produced that is available for scientific use. In general, this paper is well-motivated. The discussion on each metric is also helpful for readers. I recommend publishing this paper after minor revision.

Specific comments:

1. The metrics of OHC26 and OHC20 differ from conventional definitions for the outcropping problem. It may be better to tune the definitions (e.g., 1.5℃ as the threshold) of new OHC26 and OHC20 to maintain similar amplitudes with convectional definitions in the warm pool. Alternatively, some discussion is also beneficial about how much the new definitions overestimate or underestimate the OHC compared with the conventional definitions.

2. Three different vertically-averaged temperature metrics are introduced in this study. These metrics rely on the definition of the mixed layer depth. Plenty of papers have discussed the differences between these definitions of mixed layer depth. I think some references may be helpful for readers to understand these thermal metrics.

Line 132: The definition should be specified by providing the equation for calculation.

Lines 172-174. The citation may be misplaced. It is weird to say something like “[1] further described ...”

Line 177: It is vague to say “calculate the mixing depth directly from the ocean soundings” as it seems similar to the first method. I’m not sure whether the term ocean sounding is correct, which may be referred to the ocean temperature profile.

Line 194, For some cases with strong surface cooling, the SST may be lower than subsurface temperature. So the temperature change may be specified as temperature decrease to avoid confusion in these cases.

Line 196, “For our Sept. 15, 2010 case”, remove “our”

Line 257-259, missed a parenthesis.

Reviewer 2 Report

The authors describe the formulation of a 17-year data set, based primarily on NCODA analysis fields and augmented by a variety of different observations types when and where feasible.  Using this data set, the authors define and derive seven metrics to describe upper ocean heat content.  They compare, contrast, and provide examples of each and also identify where both archived and real-time metrics are available.  

The article is well written, the topics presented are scientifically sound, and the references are appropriate. There is precedent for the approach. While the data set presented here does not extend the existing body of scientific knowledge, the creation and availability of such a data set is of tremendous value and should facilitate many advances in the future.  Describing the development and availability of this ocean data set in a meteorology journal is appropriate, as it is likely to be heavily utilized in tropical cyclone research, specifically in studies related to intensity, structure, and the ocean response to these storms.

I recommend acceptance upon completion of the following very minor revisions:

1) Line 124: Suggest alternate verbiage for “we prefer” in “equatorward of where we prefer the metric be defined for TC studies…”.  Perhaps something like “equatorward of that often preferred for TC studies….”.

2) Ensure units adhere to the journal format and are consistent throughout the text.  Examples include:

a) Line 217: Adjust “kg/m3” to “kg m-3” as in line 118.

b) Line 269: Shift “ms-1” to “m s-1”.

c) Line 295; Figure 13 caption:  Shift “Kj/cm**2” to “kJ cm-2” as in line 288.  

d) Line 323: Shift “cm-2”  to “cm-2”.

e) Line 324: Standardize spacing for units of temperature; e.g. “0.5 ° C” in line 324 or “26° C” in line 358.

4) Line 256: Suggest shifting “stratified mixing” to “stratified, mixing”.

5) Lines 256-260:  Clarify the sentence beginning “Higher values of OHC26…”.  There is a closed paren where a comma might belong.  The definition of rapid intensification is also awkward and should be reworded.

6) Line 287: Change “30 degrees’ latitude” to “30° latitude”.

7) Line 276: The comma after “reliability” seems awkward.  Perhaps “reliability, (as determined by Pierce and Brier skill scores)”, but other options are possible.

8) Line 279:  Perhaps “analysis” should read “analyses”.

9) Ensure references are ordered properly.  [29] is in line 54, immediately after [2] in line 53 and before [3] in line 70.  Also, [16] first appears in line 332, after [28] in line 321.

10) Line 315:  To access this web address, I needed to use https:// instead of http://.  

11) Figure 5: It is challenging to distinguish the various shades of red.

12) Ensure dates consistently follow the format for this journal.  For example, in lines 138 and 180, the date is written as “September 15, 2010”, while in lines 105 and 196, it is written as “Sept. 15, 2010”, and in the captions for Figures 4 and 5 it is written as “15 September 2010”.

Author Response

Thanks for the review, please see attachment.

Reviewer 3 Report

Review of "An upper ocean thermal field metrics dataset" by Sampson et al.

This manuscript presents some global metrics of upper-ocean temperature and heat content. The data are made available publicly in real-time through a public website. This is potentially a valuable resource for the TC-ocean community.

While the idea is good, I found the selection of metrics and discussion of strengths/weaknesses to be overly simplistic. There has already been a lot of published work assessing the differences/strengths/weaknesses of ocean heat content vs. vertically-averaged temperature vs. SST. If the authors simply want to calculate these in real-time and make them available to the public, I don't think a paper needs to be published to describe it. The authors introduce some different parameters (heat content to 20C isotherm and different mixed layer temperatures), but the motivation for doing that is not clear, and there is very little discussion of strengths/weaknesses (why should someone use mixed layer-averaged temperature instead of SST?). For the manuscript to be publishable, it must be expanded/revised to other metrics with clearer and more in-depth justification/analysis of why they are important. The other option is to go simple and just provide SST, OHC, and T100, in which case a full research paper isn't justified.

Detailed comments:

Section 2: From the description, it's not completely clear what the NCODA data is. There's no ocean model mentioned, so are you using a purely observational analysis? Is the data assimilated into a model and you're using that model output? If so, are you using an analysis or a forecast? It seems like it's an analysis. Which ocean model is used for the assimilation?

Lines 96-109: Is it necessary to discuss and show SST in a figure? Every reader should know what the mean global SST distribution looks like. I recommend only showing SST in comparison to other metrics (shown together in the same figure) or difference plots to contrast other metrics with SST.

Lines 130-137: Why did you choose a temperature difference of 1C? Is it to make the maximum negative values of OHC similar in magnitude to the maximum positive values, or is it arbitrary? Ideally you could scale it so it's approximately linear: for OHC=-100 a TC is about twice as likely to decrease in intensity (or the magnitude of that decrease is about twice as large on averqge) compared to OHC=-50. Another method would be to use a fixed depth for the temperature integration (maybe 100 m) to avoid the 26C/20C issue.

Lines 161-162: I don't see a separate maximum in the subtropical western North Pacific in Fig. 5. Please explain. I recommend combining Figs. 2-5 into two figures: one with (a) depth of 26C isotherm, (b) depth of 20C isotherm, (c) difference between (a) and (b). The other would have (a) OHC calculated using 26C, (b) OHC using 20C, and (c) difference between (a) and (b).

Line 183: I don't see any obvious differences between Fig. 6 and Figs. 3 and 5. Please explain. The advantages/disadvantages of OHC/TCHP were discussed in some detail in ref. (20).

Lines 191-192: What do you mean by 'barrier layer?' Normally it refers to the depth range between the base of the density-based mixed layer and the base of the temperature-based mixed layer. However, there is not always a barrier layer. Please clarify.

I don't understand why you include the mixed layer-averaged temperatures as metrics. They will always be very similar to SST and seem unnecssary. They differ from SST based on the criterion used to define the mixed layer, the depth of the mixed layer, and the temperature stratification at the base of the mixed layer. A larger criterion will result in lower mixed layer temperature relative to SST. A shallower mixed layer will result in a lower mixed layer temperature compared to a thicker mixed layer, everything else equal (because the depths where temperature is decreasing make up a larger fraction of the total depth of the mixed layer, resulting in a colder mixed layer-averaged temperature). It's unclear how the mangitude of temperature stratification will affect the mixed layer-averaged temperature. Based on these considerations, what is the purpose of including mixed layer-averaged temperature as a metric? There are better ways to define a variable depth for calculating the temperature average (e.g., Balaguru et al. 2015, GRL). Plots of differences between SST and the various mixed layer-averaged temperatures would be good to see to determine whether there's any value in them.

Section 3.1.7: Again, 'barrier layer' normally refers to salinity stratification, but I think you're using it to describe density stratification. It's a little confusing. The method in this section is just another way to estimate mixed layer depth, so it's not surprising that the results are very similar to the other mixed layer-averaged temperatures. There are better ways to incorporate salinity into the vertically averaged temperature, by estimating its tendency to reduce mixing and limit mixed layer deepening as a TC passes over. 

Why is the depth of max. stability so patchy in Fig. 11? That tells me it might not be a very reliable metric. Better would be a depth that takes into account the depth of TC-induced mixing, which depends on TC intensity, translation speed, etc., or a depth that is based on an available potential energy threshold (vertical integral of rho*z to a depth h, minus *h, where <> is a vertical average from the surface to h). It's another way to define MLD.

I strongly recommend making the data available to the public in netcdf format.

Author Response

Thanks for reviewing, please see attachment.

Round 2

Reviewer 3 Report

The authors have not addressed/incorporated many of my comments from the first review. However, I see the value in the manuscript as a data paper that presents real-time ocean products. In that context, new results or a new dataset (with unique metrics) are not needed. As a potential user of the dataset, I'm disappointed that it will not be available in a common format like netcdf, but if the target audience is one that is comfortable with GRIB, it's understandable, and it's your choice as dataset creator. Netcdf is much more commonly used by the research community, which I think could benefit from your dataset.